# CAN COPYRIGHT BE REDUCED TO PRIVACY?

## ABSTRACT

There is a growing concern that generative AI models may generate outputs that closely resemble the copyrighted input content used for their training. This worry has intensified as the quality and complexity of generative models have immensely improved, and the availability of extensive datasets containing copyrighted material has expanded. Researchers are actively exploring strategies to mitigate the risk of producing infringing samples, and a recent line of work suggests employing techniques such as differential privacy and other forms of algorithmic stability to safeguard copyrighted content. In this work, we examine whether algorithmic stability techniques such as differential privacy are suitable to ensure the responsible use of generative models without inadvertently violating copyright laws. We argue that there are fundamental differences between privacy and copyright that should not be overlooked. In particular, we highlight that although algorithmic stability may be perceived as a practical tool to detect copying, it does not necessarily equate to copyright protection. Therefore, if it is adopted as a standard for copyright infringement, it may undermine the intended purposes of copyright law

## 1 INTRODUCTION

Recent advancements in Machine Learning have sparked a wave of new possibilities and applications that could potentially transform various aspects of our daily lives and revolutionize numerous professions through automation. However, training such algorithms heavily relies on extensive content, either annotated or generated by individuals who might be effected by these algorithms. Consequently, determining when and how content can be used within this framework without infringing upon individuals' legal rights have become a pressing challenge. One area where this issue arises prominently is in the operation of generative models, which takes as input human-produced content, which is often copyrighted, and are expected to generate "similar" content. For instance, consider a machine that observes images and is tasked with producing new images that resemble the input. In this context, the fundamental question arises:

*When does the content generated by a machine (output content) infringe copyright in the training set (input content).*

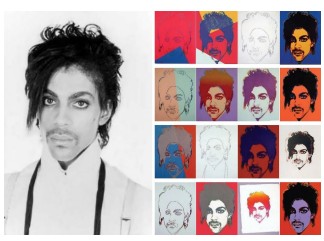

Figure 1

This question is not purely theoretical, as various aspects of this problem have already become subjects of legal disputes in recent years. In 2022 a class action was filed against Microsoft, GitHub, and OpenAI, claiming that their code-generating systems, Codex and Copilot, infringed copyright in the licensed code that the system was allegedly trained on Dvs.G (2022). Similarly, in another class action, against Stable Diffusion, Midjourney, and DeviantArt, plaintiffs argue that by training their system on web-scraped images, the defendant infringe the rights of millions of artists AVs.S (2023). Allegedly, the images produced by these systems, in response to prompts provided by the systems' users, are derived solely from the training images, which belong to plaintiffs, and, as such, are regarded as unauthorized derivative works of the plaintiffs' images and constitute a copyright infringement.

A preliminary question is whether it is lawful to make use of copyrighted content in the course of training Lemley & Casey (2020); Grimmelmann (2015); Legislation & Law) (2022). There are com-

pelling arguments to suggest that such intermediary copying might be considered fair use Lemley & Casey (2020). For example, Google's Book Search Project—entailing the mass digitization of copyrighted books from university library collections to create a searchable database of millions of books—was held by US courts to be fair use Google (2015). Then, there is a claim that generative models reproduce protected copyright expressions from the input content on which the model was trained. However, to claim that the output of a generative model infringes her copyright, a plaintiff must prove not only that the model had access to her copyrighted work, but also that the alleged copy is substantially similar to her original work Svs.M (1977); Bvs.S (1992)

But, trying to identify what constitutes "substantial similarity", as well as *alleged copies* remains a challenge. Recent studies have proposed measurable metrics to quantify copyright infringement Vyas et al. (2023); Bousquet et al. (2020); Scheffler et al. (2022); Carlini et al. (2023). One approach, Vyas et al. (2023); Bousquet et al. (2020) asserts that copyright is not infringed if it is reasonable for the machine to output the content even when it had no access to the protected content. The argument can be illustrated as follows: Suppose that Alice outputs content A and Bob claims it plagiarizes content B. Alice may argue that she never saw content B, and would reason that this means she did not infringe Bob's copyright. However, since Alice must observe some content, a second line of defense could be that "**had** she never saw B" she would still be likely to produce A. The above argument was exemplified by Bousquet et al. (2020) that interprets differential-privacy in the above manner. Subsequently, Vyas et al. (2023) presented a certain generalization, in the form of a *near free access* (NAF) notion that can potentially allow a more versatile notion of copyright protection.

As far as differential privacy, certain traits of copyright law make it challenging to reduce the problem to a question of privacy. To begin with, an important element of copyright law in the United States is that it has a utilitarian rationale, seeking to promote the creation and deployment of creative works CONST (()); Mvs.S (1954). It is important, then, that any interpretation of copyright, or for that matter any quantifiable measure for copyright, will be aligned with these objectives. In particular, while the law delineates a set of exclusive rights to the creators of original expressions, it must ensure sufficient creative space for current and future creators Samuelson (2016). As such, certain issues already distinguish copyright law from privacy defined by criteria such as algorithmic stability. Copyright is limited in time, and once protection has expired the work enters the public domain and is free for all to use without authorization Litman (1990). This issue, though, can be modeled by a distinction between private and public data (or protected and non-protected data). However, more importantly, to achieve its goal, copyright law excludes certain subject matter from protection (e.g. ideas, methods of operation, facts), since they are regarded as raw materials for cultural expression. In contrast, requirements such as privacy protect content and not expression, which in turn can be misaligned with the original objectives of copyright law.

Another distinction from privacy is that copyright law encourages the use of copyrighted materials by exempting certain types of transformative uses, such as quotations, parodies, and some other fair uses such as learning and research Netanel (2011). The fair use doctrine serves as a check on copyright, to ensure it does not stifle the very creativity copyright law seeks to foster. Fair use is also considered one of the safety valves which allows copyright protection to coexist with freedom of expression Netanel (2008).

In this study we initiate a discussion about the challenges involved in providing a rigorous definition that captures the concept of copyright. We commence with a technical discussion, comparing different proposed notions of copyright (in particular, differential privacy and NAF) and examine their close connection to algorithmic stability. Subsequently, we argue that any approach following this line of reasoning encounters significant obstacles in modeling copyright as understood within the legal context. In more detail, we argue that algorithmic stability strategies fail to account for some key features of copyright law that intend to preserve copyright delicate balance. We identify several major gaps between algorithmic stability strategies and copyright doctrine, demonstrating why applying such strategies may fail to account for essential copyright concepts. Therefore, we argue, that if algorithmic stability techniques are adopted as a standard for copyright infringement, they may undermine the intended objectives of copyright law. We further propose a different approach to using quantified measures in copyright disputes, to better serve and reconcile copyright trade-offs.

## 1.1 RELATED WORK

A growing number of researchers in recent years explore how to address legal problems by applying theories and methods of computer science. This literature seeks to narrow the gap between the vague and abstract concepts used by law and mathematical models, and to offer more rigor, coherent and scalable definitions for issues such as privacy Dwork & Feldman (2018), or fairness and discrimination Dwork et al. (2012); Kairouz et al. (2021) In the context of generative models Carlini et al. (2023); Haim et al. (2022) ] explore whether generative diffusion models memorize protected works that appeared in the models' training set. This can be considered as a preliminary question to the problem of copyright. However, as discussed, memorization of the input content does not necessarily equate to copyright infringement, and we must therefore consider other measurable metrics and quantified measures for copyright key limiting concepts.

There is also active and thought-provoking discussion on how ML technologies are reshaping our understanding of copyright within the realm of law. Asay (2020) explores the question of whether AI system outputs should be subject to copyright. Our focus, though, is on the legitimacy of using copyrighted material by AI. Additionally, as discussed, Grimmelmann (2015); Lemley & Casey (2020) explore the implications of copyright law for literary machines that extract content and manage databases of information.

The works of Bousquet et al. (2020); Vyas et al. (2023) which rely on privacy/privacy-like notions, is the main focus of our work. An alternative approach taken by Scheffler et al. (2022) proposes a framework to test substantial similarity by comparing Kolmogorov-Levin complexity with and without access to the original copyright work. Beyond algorithmic challenges, to apply a substantial similarity test. However, one has to provide a distinction between protected expressions and non-protected ideas, which may in some cases be the crucial challenge that we might want to solve. Another approach Franceschelli & Musolesi (2022) suggests to use generative learning techniques to assess creativity. Henderson et al. (2023) seek to develop strategies to be applied to generative models to ensure they satisfy the same fair use standard as in human discretion. The application of this solution may not be possible, though, in cases where little to no open source or fair use data is readily available.

## 2 ALGORITHMIC STABILITY AS A SURROGATE FOR COPYRIGHT

In this section, our focus is to introduce and discuss two notions of algorithmic stability: near-access-freeness (NAF) and differential privacy (DP); these two notions were specifically investigated in the realm of training methods aimed at safeguarding copyrighted data.

Both NAF and DP adhere to a shared form of stability: they ensure that the resulting model, denoted as $q$, satisfies a safety condition with respect to each copyrighted data instance, denoted as $c$. This safety condition guarantees the existence of a "safe model", denoted by $q_c$, which does not infringe the copyright of data $c$, and importantly, $q$ exhibits sufficient similarity to $q_c$. Consequently, both NAF and DP guarantee that $p$ itself does not violate the copyright of the respective data instance $c$.

Formally, we consider a standard setup of an unknown distribution $\mathcal{D}$, and a generative algorithm $A$. The algorithm $A$, gets as an input a training set of i.i.d samples $S = \{z_1, \ldots, z_m\} \in Z^m \sim D^m$, and outputs a model $p_S^A = A(S)$, which is a distribution supported on $Z$. For simplicity, we will assume here that $Z$ is a discrete finite set, but of arbitrary size. Vyas et al. (2023) consider a more general variant in which the output posterior is dependent on a "prompt" $x$, and $A$ outputs a mapping $p^{(A_S)}(\cdot|x)$ that may be regarded as a mapping from prompts to posteriors. For our purposes there is no loss in generality in assuming that $p$ is "promptless", and our results easily extend to the promptful case, by thinking of each prompt as inducing a different algorithm when we hard-code the prompt into the algortihm.

**Differential Privacy** $A$ is said to be $(\alpha, \beta)$-differentially private Dwork et al. (2006) if for every pair of input datasets $S, S'$ that differ on a single datapoint, we have that for every event $E$:

$$\mathbb{P}(A(S) \in E) \leq e^\alpha \mathbb{P}(A(S') \in E) + \beta \text{ and } \mathbb{P}(A(S') \in E) \leq e^\alpha \mathbb{P}(A(S) \in E) + \beta \quad (1)$$

The concept of privacy, viewed as a measure of copyright, can be explained as follows: Let's consider an event, denoted as $E$, which indicates that the generative model produced by $A$ violates the

copyright of a protected content item $c$. The underlying assumption is that if the model has not been trained on $c$, the occurrence of event $E$ is highly improbable. Thus, we can compare the likelihood of the event $E$ when $c$ is present in the sample $S$ with the likelihood of $E$ when $c$ is not included in a neighboring sample $S'$ (which is otherwise identical to $S$). If $A$ satisfies the condition stated in equation Eq. (1), then the likelihood of event $E$ remains extremely low, even if $c$ happened to be present once in its training set.

**Near Access Freeness**  There are several shortcomings of the notion of differential privacy that have been identified. Some of these are reiterated in Section 3. Vyas et al. (2023) proposed the notion of Near-Access Freeness (NAF) that relaxes differential privacy in several aspects. Formally, NAF (or more accurately NAF w.r.t safe function safe and $\Delta_{max}$ is defined as follows: First, we assume a mapping safe that assigns to each protected content $c$ a model $q_c$ which is considered safe in the sense that it does not breach the copyright of $c$. The function safe, for example, can assign $c$ to a model that was trained on a sample that does not contain $c$. Several safe functions have been suggested in Vyas et al. (2023).

A model $p$ is considered $\alpha$-NAF if the following inequality holds simultaneously for every protected content $c$ and every $z$:
$$p(z) \leq e^{\alpha} q_c(z). \tag{2}$$
The intuition behind NAF is very similar to the one behind DP, however there are key differences that can, in principle, help it circumvent the stringency of DP.

1. The first difference between NAF and DP is that the NAF framework allows more flexibility by picking the 'safe' function. Whereas DP is restricted to a safe model corresponding to training the learning algorithm on a neighboring sample excluding the content $c$.

2. A second difference is the fact that NAF is one sided (see Eq. (2)), in contrast with DP which is symmetric (see Eq. (1)). Note that one-sidedness is indeed more aligned with the requirement of copyright which is non-symmetric.

3. NAF makes the distinction between content-safety and model-safety Vyas et al. (2023). In more detail, the NAF notion requires that the output model is stable. This is in contrast with privacy that requires stability of the posterior distribution over the output models. In this sense the notion of NAF is more akin to *prediction differential privacy* Dwork & Feldman (2018) then to differential privacy.

4. Finally, NAF poses constraints on the model outputted by the learning algorithm (each constraint corresponds to a prespecified *safe model*). This is in contrast with privacy which does not restrict the output model, but requires stability of the posterior distributions over output models. This distinction may seem minor but it can lead to peculiarities. For example, an algorithm that is completely oblivious to its training set and that always outputs original content can still violate the requirements of NAF. To see this, imagine that our learning rule outputs a model $q$ that always generates the same content $z$ which is completely original and not similar to any protected content $c$. However, depending on the safe models $q_c$ it can be the case that the model $q$ is not similar to any of them.

The above differences, potentially, allow NAF to circumvent some of the hurdles for using DP as a notion for copyright. For example, the one-sidedness seems sufficient for copyright and may allow models that are discarded via DP. Also, the distinction between model-safety and content-safety can, for example, allow models that may memorize completely the training set as long as a content they output does not provide a proof for such memorization. Next, the fact that NAF is defined by a set of constraints, and not a property of the learning algorithm, allows one to treat breaches of Eq. (2) as soft "flagging" and not necessarily as hard constraints. This advantage is further discussed in Section 4. Finally, perhaps most distinguishable, is the possibility to use general safety functions that can capture copyright breaches more flexibly. We next discuss the implications of these refinements. We begin with the question of model safety vs. content safety in NAF and in DP.

**Model safety vs. Content safety**  Our first result is a parallel to Theorem 3.1 in Vyas et al. (2023) in the context of DP stability. Theorem 3.1 in Vyas et al. (2023) shows how to efficiently transform a given learning rule $A$ to a learning rule $B$ which is NAF-stable, provided that $A$ tends to output similar generative models when given inputs that are identically distributed. We state and prove a

similar result by replacing NAF stability with DP stability, which demonstrates that the notion of DP can be relaxed , analogously to NAF, to require only content safety under proper assumptions:

Recall that the total variation distance between any two distributions is defined as: $\|q_1 - q_2\| = \frac{1}{2}\sum|q_1(x) - q_2(x)| = \sup_E (q_1(E) - q_2(E))$,

**Proposition 1.** *Let $A$ be an algorithm mapping samples $S$ to models $q_S^A$ such that*

$$\mathop{\mathbb{E}}_{S_1, S_2}\left[\|q_{S_1}^A - q_{S_2}^A\|\right] \leq \alpha,$$

*where $S_1, S_2 \sim D^m$ are two independent samples. Then, there exist an $(\epsilon, \delta)$ DP algorithm $B$ that receives a sample $S_B \sim D^{m_{priv}}$ such that if $m_{priv} = \tilde{O}\left(\frac{m}{\eta\epsilon}\log 1/\delta\right)$ and $S_A \sim D^m$ then:*

$$\mathop{\mathbb{E}}_{S_A, S_B}\left[\|\mathbb{E}[q_{S_B}^B] - q_{S_A}^A\|\right] \leq \frac{2\alpha}{1+\alpha} + O(\eta).$$

*Where the expectation within, is taken over the randomness of $B$.*

The premise in the above theorem is identical to that in Theorem 3.1 in Vyas et al. (2023) and captures the property that $A$ provides similar outputs on identically distributed inputs. The obtained algorithm $B$ is DP-stable and at the same time it has a similar functionality like $A$ in sense that its output model $q^B$ generates content $z$ which in expectation is distributed like contents generated by $q^A$.

**Safety functions**   We now turn to a discussion on the potential behind the use of different safety functions. The crucial point (which we discuss in great detail in Section 3 below) is that a satisfactory "copyright definition" *must* allow algorithms to be highly influenced, even by their input content which is *protected*. This reveals a stark contrast with algorithmic stability: it is easy to see that DP does not allow such influence. Indeed, the whole philosophy behind privacy is that a model is "safe" if it did not observe the private example (and in particular was not influenced by it).

This raises the question of whether the greater flexibility of the NAF model can provide better aligned notions of safety. In fact, if it is allowed to be influenced by protected data, one might even want to consider safe models that have *intentionally* observed a certain content and derived out of it the derivatives that are not protected.

The next result, though, shows that there is a *no free lunch* phenomenon. For every protected content $c$, we can either only consider safe models that observed $c$ and are influenced by it, or only safe models that *never* observed it and were *not* influenced by it. In other words, if a protected content $c$ influenced its safe model $q_c$ then it must influence all safe models $q_{c'}$ for all protected contents $c'$. We further elaborate on the implication of this result in Section 4.

Below, $q_1$ and $q_2$ should be thought of as safe models, and $p$ as the model outputted by the NAF learning algorithm. (So, in particular $p$ should satisfy Eq. (2) w.r.t $q_1$ and $q_2$.) This result complements Theorem 3.1 in Vyas et al. (2023) which shows that NAF can be satisfied in the sharded-safety setting when the two safe models are close in total-variation. The proof is left to Appendix A.1.

**Proposition 2.** *Let $q_1$ and $q_2$ be two distributions such that $\|q_1 - q_2\| \geq \alpha$, then for any distribution $p$ we have that for some $z$:*

$$p(z) \geq \frac{1}{2(1-\alpha)}\min\{q_1(z), q_2(z)\},$$

## 3   THE GAP BETWEEN ALGORITHMIC STABILITY AND COPYRIGHT

So far, we have provided a technical comparison between existing notions in the CS literature aimed at provable copyright protection. While the technical notion of privacy may seem closely related, as observed through NAF, there are differences and there is room for more refined definitions that may capture these essential differences. While algorithmic stability approaches hold promise in helping courts assess copyright infringement cases (an issue we further discuss in Section 4), as we will show next, they cannot serve as a definitive test for copyright infringement. In order to see that, we next discuss the issue of copyright from a legal perspective. From this perspective, formal algorithmic stability approaches are both overly inclusive and overly exclusive. Consequently, we will organize this section based on these challenges.

### 3.1 OVER-INCLUSIVENESS

Here we focus on a concern that algorithmic stability approaches may filter out lawful output content that does not infringe copyright in the input content. Because non-infringing output content is lawful, employing algorithmic stability approaches as filters to generative models may needlessly limit their production capabilities, and, thereby, undermine the ultimate objectives of copyright law. Copyright law intends to foster the creation of original works of authorship by securing incentives to authors and, at the same time, ensuring the freedom of current and future authors to use and build upon existing works. The law derives from the U.S. Constitutional authority: "To promote the Progress of Science and useful Arts, by securing for limited Times to Authors and Inventors the exclusive Right to their respective Writings and Discoveries." CONST (()

However, the goal of promoting progress is often at odds with granting unlimited control over copyrighted materials. This is why copyright law sets fundamental limits on the rights granted to authors. Promoting progress is inconsistent with unrestricted right to prevent every unauthorized use , because creators and creative processes are embedded in cultural contexts. The creative process often requires ongoing interactions with preexisting materials, whether through learning and research, engagement with prior art to generating new interpretation, or the use of a shared cultural language and applying existing styles to make works of authorship more comprehensible. Consequently, the utilization of copyrighted materials becomes a crucial input in any creative process Cohen (2012); Elkin-Koren (1996).

For this reason, unlike the mandate of the algorithmic stability approaches, copyright law does not require that an output content will not draw at all on an input content to be lawful and non-infringing. On the contrary. There are many cases where copyright law explicitly allows for an output content to heavily draw on the input content without raising infringement concerns. In such cases, allowing an input content to impact an output content is not only something that copyright law permits, it is something that copyright law encourages. Doing so, as Jessica Litman put it: "is not parasitism; it is the essence of authorship." Litman (1990)

Copyright law allows an output content to substantially draw on an input content in three main cases, which we next explore: (1) When the input content is in the public domain, (2) When the input content is copyrighted but incorporates aspects that are excluded from copyright protection, and (3) When the use of the protected aspects of the input content are lawful.

**When an input content is in the public domain**   An input content may be unprotected because its copyright term has lapsed. Copyrights are limited in duration (though relatively a long duration, which in most countries will last life of the author plus seventy years). Once the copyright term expires, an input content enters the public domain and could freely be used and impact an output content without risking copyright infringement Litman (1990). Public domain materials may also consist of anything that is not at all copyrightable, such as natural resources. For instance, if two photographers are taking pictures of the same person, some similarity between those pictures is likely due to the way this person looks, which is in the public domain. Other elements such as an original composition, or the choices made regarding lighting conditions and the exposure settings used in capturing the photograph, might be considered copyrighted expression. If the generative model only makes use of the former in the output content, it may not constitute an infringement.

**When an input content incorporates unprotected aspects**   Input contents with a valid copyright term, enjoys "full" legal protection, but it too is limited in scope. As provided by the copyright statute, "[i]n no case does copyright protection for an original work of authorship extend to any idea, procedure, process, system, method of operation, concept, principle, or discovery, regardless of the form in which it is described, explained, illustrated, or embodied in such work." U.S.C (2006). By this principle, an output content may substantially draw on an input content without infringing copyright in the latter, as long as such taking is limited to the input's content unprotected elements.

- **Procedures, processes, systems and methods of operation** Copyright protection does not extend to "useful" or "functional" aspects of copyrighted works such as procedures, systems, and methods of operation. These aspects of an input content are freely accessible for an output content to draw upon. . For example, in the seminal case of Baker vs. Selden, the Supreme Court allowed Baker to create a book covering an improved book-keeping system while drawing heavily on the

charts, examples, and descriptions used in Selden's book without infringing Selden's copyright Bvs.S (1879). As the court explained, these aspects that Baker took from Selden's work are functional methods of operations and as such are not within the domain of copyright law. Similarly, in Lotus v. Borland, the United States Court of Appeals for the First Circuit allowed Borland to copy Lotus's menu command hierarchy for its spreadsheet program, Lotus 1-2-3. The court ruled that Lotus menu command hierarchy was not copyrightable because they form methods of operation L.vs.B (1996) - Consequently, if a generative model simply extracts procedures, processes, systems and methods from the training set it may not infringe copyright.

- **Ideas** Copyright protection is limited to concrete "expressions" and does not cover abstract "ideas." Thus, in Nicholas v. Universal, the United States Court of Appeals for the Second Circuit allowed Universal to incorporate many aspects of Anne Nichols' play Abie's Irish Rose, in their film The Cohens and Kellys vs. U (1930). The court explained that the narratives and characters that Universal used ("a quarrel between a Jewish and an Irish father, the marriage of their children, the birth of grandchildren and a reconciliation"), were "too generalized an abstraction from what she wrote. . . [and, as such]. . . only a part of her [unprotected] 'ideas.'" vs. U (1930) When a generative model simply extract ideas from copyrighted materials, rather than replicating expressive content from their training data, it does not trigger copyright infringement.

- **Facts** Copyright protection also does not extend to facts. For example, in Nash v. C.B.S., the court ruled that C.B.S. could draw heavily from Jay Robert Nash's books without infringing his copyright N.vs.C (1990). As the court explained, the hypotheses that Nash rose speculating the capture of the gangster John Dillinger and the evidence he gathered (such as the physical differences between Dillinger and the corpse, the planted fingerprints, and photographs of Dillinger and other gangsters in the 1930s) were all unprotected facts that Nash could not legally appropriate. Consequently, generative models which simply memorize facts do not infringe copyright law.

**When the use of the protected aspects of the input content was lawful** Even when the protected elements of an input content ("expressions" rather than the "ideas") are impacting an output content, such impact may be legally permissible. There are two main categories of lawful uses: de minimis copying and fair use.

- **De minimis copying** Copyright law allows de minimis copying of protected expression, namely the coping of an insignificant amount that has no substantial impact on the rights of the copyright owner or their economic value. In a similar way, "[w]ords and short phrases, such as names, titles, and slogans, are uncopyrightable."Office (2021). However, de minimis coping of protected expression may be unlawful if it captures the heart of the work Hvs.R (1985). For example, phrases like "E.T. Phone Home." Uvs.K (1982)

- **Fair Use** Copyright law also allows copying of protected expression if it qualifies as fair use. The U.S. fair use doctrine, as codified in § 107 of the U.S. Copyright Act of 1976, is yet another legal standard to carve out an exception for an otherwise infringing use after weighing a set of four statutory factors. The four statutory factors are: (1) the purpose and character of the use, including whether such use is of a commercial nature or is for nonprofit educational purposes; (2) the nature of the copyrighted work; (3) the amount and substantiality of the portion used in relation to the copyrighted work as a whole; and (4) the effect of the use upon the potential market for or value of the copyrighted work U.S.C (2006).

Importantly, the fair use claimant need not satisfy each factor in order for the use to qualify as fair use Cvs.A (1994). Nor are the four factors meant to set out some kind of mathematical equation whereby, if at least three factors favor or disfavor fair use, that determines the result Netanel (2011). Rather, the factors serve as guidelines for holistic, case-by-case decision. In that vein, in its preamble paragraph, § 107 provides a list of several examples of the types of uses that can qualify as fair use. The examples, which include "criticism, comment, news reporting, teaching (including multiple copies for classroom use), scholarship, [and] research,"U.S.C (2006) are often thought to be favored uses for qualifying for fair use. Importantly, however, the list of favored uses is not dispositive. Rather, fair use's open-ended framework imposes no limits on the types of uses that courts may determine are "fair" Cvs.A (1994).

When the factors strongly favour a finding of fair use, even output contents that are heavily impacted by copyrighted input contents may be excused from copyright infringement. For example, in

Campbell v. Acuff-Rose, although the rap music group 2 Live Crew copied significant portions of lyrics and sound from Roy Orbison's familiar rock ballad "Oh, Pretty Woman" Cvs.A (1994). The Supreme Court denied liability in this case, based on the premise that the 2 Live Crew's derivative work was considered a "parody" of Orbison's original work, and, therefore, constituted fair use. Similarly, in The Authors Guild v. Google, the court defended Googles' mass digitization of millions of copyrighted books to create a searchable online database as fair use, because it considered Google's venture to be socially desirable Google (2015) as explained by Sag (2018), concluding that the copying of expressive works for non-expressive purposes should not be counted as a copyright infringement.

## 3.2 Overly Exclusiveness

Algorithmic stability approaches are under-exhaustive because they might fail to filter out unlawful output content that infringes copyright in the input content. As explained, algorithmic stability approaches find infringement only when the output content heavily draws on an input content. The law of copyright infringement, however, is not so narrow. Copyright law only requires that the output content will heavily draw on the protected expression that originated from an input content to find infringement. Such expression need not come from the input content itself; it may come from other sources including copies, derivatives or snippets of the original content Lee et al. (2023).

To illustrate this point, consider the fact pattern in the U.S. Supreme Court case Warhol vs. Goldsmith Wvs.G. In that case, the portrait photographer Lynn Goldsmith accused Andy Warhol of infringing copyrights in a photograph she took of the American singer Prince. Goldsmith authorized Warhol to use her photograph as an "artistic reference" for creating a single derivative illustration (see Fig. 1, bottom right most picture). Still, she did not approve nor imagine that Warhol had, in fact, made 16 different derivatives from the original photograph. Warhol's collection of Prince portraits, also known as the Prince series , is depicted in Fig. 1, right side.

For our purposes, assume the Prince Series' portraits served as input for a generative machine. If the machine's output content draws heavily on Goldsmith's protected expression that is baked into the Prince Series' portraits, then the output content may infringe Goldsmith's copyright in original photograph (Fig. 1 , left side), ), even if the machine did not have access to Goldsmith's original photograph. Moreover, this risk will not be eliminated even if the Supreme Court were deciding that the Prince Series' portraits themselves are non-infringing because they constitute fair use.

Putting it simply, copying from a derivative work—whether authorized by the copyright owner or not— may infringe copyright in the original work on which the derivative work is based. This situation is prevalent in copyright practice, especially in music. In modern music copyright cases, plaintiffs usually show access to the original copyrighted work (musical composition) by showing access to a derivative work of that original work (sound recording). Plaintiffs are not required to demonstrate that the defendants also had access to the original sheet music nor that they could actually read musical notes.

Lastly, output content can also infringe copyright in input content by accessing parts or snippets of the input content even without accessing the input content in its entirety. This concern was raised recently in The Authors Guild v. Google, a case dealing with the legality of the Google Book Search Library Partner project Google (2015). As part of this project, Google scanned and entered many copyrighted books into their searchable database but only provided "snippet views" of the scanned pages in search results to their users. The plaintiff in the case argued that Google facilitated copyright infringement by allowing users to aggregate different snippets and reconstruct infringing copies of their original works. The court ended up dismissing this claim, but only because Google took affirmative steps to prevent such reconstruction by limiting the number of available snippets and by blacklisting certain pages.

## 4 Discussion

Algorithmic stability approaches, when used to establish a proof of copyright infringement are either too strict or too lenient from a legal perspective. Due to this misfit, applying algorithmic stability approaches as filters for generative models is likely to distort the delicate balance that copyright law aims to achieve between economic incentives and access to creative works.

The purpose of this article is to illuminate this misfit. This is not to say that algorithmic approaches in general and algorithmic stability approaches in particular have no value to the legal profession. Quite the opposite. Computer science methodologies bring a significant benefit to the judicial table: the capability to process large volumes of information and assist policymakers in making more informed decisions. Many areas in law involve applying murky "standards" as oppose to rigid "rules." Kaplow (1992). As discussed, copyright law makes extensive use of legal standards, such as idea/expression distinction, or fair use principles, to restrict the scope of protection accorded to copyrighted works. Consequently, copyright infringement cannot be boiled down to a binary computational test.

The true value of computer science methodologies to the legal profession is not necessarily to convert murky standards into rigid rules (e.g., by constructing a definitive binary test for copyright infringement), but, instead, to make legal standards less murky. A rich body of scholarship explores the ills of vaguely-defined legal standards, especially in the context of intellectual property Parchomovsky & Stein (2009); Benkler (1999); Samuelson (1996); Gibson (2006); Menell & Meurer (2013) Algorithmic stability approaches, if applied with caution, may introduce new quantifiable methods for applying legal standards in a clearer and more predictable manner. Such methods could help measure vague legal concepts such as "fairness" "privacy," and, in the copyright context—"originality", and at the same time facilitate the ongoing development of legal and social norms Hacohen & Elkin-Koren (2024). At the same time, to ensure these methods are beneficial, it is important to acknowledge the limitations of applying algorithmic stability approaches to copyright.

**Stability is not safe**   The NAF framework, that allows a rich class of safety functions, has the potential to circumvent some of the challenges presented, but may still be limited and we now wish to discuss this in further details.

To utilize the NAF framework, the first basic question one needs to address is *Given a protected content $c$ how should we choose the safe model* $\mathrm{safe}(c)$*?* It seems natural to include models that are not heavily influenced by $c$ since otherwise this might allow copyright breaching. However, such choice of $\mathrm{safe}(c)$ leads to the discussed limitations encountered by algorithmic-stability approaches such as DP. It is true that some aspects, such as content safety vs. model safety, can be better aligned through the definition of NAF but also, as Proposition 1 shows, through variants of DP. Overall, there is room, then, to further investigate the different possible models for copyright, within such an approach, but we should take into account the limitations presented in Section 3.

Perhaps a more exciting application of NAF, then, is to consider notions of safety that allow some influence by $c$. e.g. to enable generating parodies, fair-use, de minimis copying, etc. We consider then safety functions that now *do* have access to $c$, and exploit this access to validate that only allowed influence happens. Here we face a different challenge. Suppose that $q_c$ is such a safe model for content $c$. Suppose, also, that $q_{c'}$ is another safe model for content $c'$. If $q_{c'}$ and $q_c$ are far away, then Proposition 1 shows that there is no hope to output a NAF model. But even if $q_c$ and $q_{c'}$ are not far away, but suppose that $q_{c'}$ ignores content $c$, then for any content $z$ that is influenced by $c$ we may assume that:

$$q_c(z) \gg q_{c'}(z).$$

But, if $p$ is a NAF model, we must also have due to Eq. (2) with respect to $c'$ and $z$:

$$q_c(z) \gg p(z).$$

In other words, the NAF model censors permissible content $z$ even though it is safe. This happens because $z$ is an improbable event in model $q_{c'}$. Not because $z$ breaches copyright of $c'$ but because it is influenced by $c$, and content that is influenced by $c$ is discarded by safe models that had no access to $c$. It follows, then, that all safe models must treat protected content in a similar manner, and $q_{c'}$ must also be influenced by $c$ if we expect the NAF model to make any use of it. Hence, it is unclear if a more refined notion of $\mathrm{safe}$ may help circumvent the hurdles of applying the privacy approach for establishing a copyright infringement. This suggests, though, to perhaps consider a relaxed variant of NAF in which a content is discarded by a safe model only when certain links between the protected content and the generated content are established.

It seems, then, that an algorithmic approach that assist jurists in understanding such links between existing works of authorship, study their hidden interconnection, and quantify their originality can hold great promise. From this perspective, originality is evaluated by the semantic distance between a measured expressive work and similar materials found in the corpus of pre-existing expressions. Research in this area is just beginning but holds a great promise for the copyright system.

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

# A  PROOFS

## A.1  PROOF OF PROPOSITION 2

Suppose that

$$\|q_1 - q_2\| \geq \alpha.$$

In particular there exists an event $E$ such that:

$$q_2(E) \leq q_1(E) - \alpha \leq 1 - \alpha.$$

Let $p$ be some distribution. We assume that $p(E) \geq 1/2$ (otherwise, replace $E$ with its complement and $q_1$ and $q_2$ replace roles). Thus, we have that:

$$p(E) \geq \frac{1}{2} \geq \frac{1}{2(1-\alpha)} q_2(E).$$

In particular, for some $z \in E$, the result follows.

## A.2  PROOF OF PROPOSITION 1

The proof relies on a coupling Lemma, taken from Angel & Spinka (2019). Recall that, given a collection of distribution measures $Q$, a coupling can be thought of as a collection of random variables $X = (X_q)_{q \in Q}$, whose marginal distributions are given by $q$. I.e. $\mathbb{P}(X_q = x) = q(x)$:

**Lemma 1** (A special case of Thm 2 in Angel & Spinka (2019)). *Let $Q$ be the collection of all posteriors over a finite domain $\mathcal{X}$[1]. There exists a coupling such that for every $q, q' \in Q$:*

$$\mathbb{P}(X_q \neq X_{q'}) \leq \frac{2\|q - q'\|}{1 + \|q - q'\|}.$$

The second Lemma we rely on is a private heavy hitter mechanism, described as follows:

**Lemma 2** (Korolova et al. (2009); Bun et al. (2016)). *Let $Z$ be a finite data domain. For some*

$$k \geq \Omega\left(\frac{\log 1/\eta\beta\delta}{\eta\epsilon}\right),$$

*there exists an $(\epsilon, \delta)$-DP algorithm* hist, *such that with probability $(1 - \beta)$ on an inputs $S = \{z_1, \ldots, z_k\}$ outputs a mapping $a \in [0, 1]^Z$, such that, for every $z \in Z$,*

$$|a(z) - freq_S(z)| \leq \eta.$$

*In particular, if $freq_S(z) > 0$, then $a(z) > 0$.*

Where we denote by $\text{freq}_S(z) = \frac{|i : z_i = z|}{|S|}$.

We next move on to prove the claim. Let $X$ be the coupling from Lemma 1. Our private algorithm works as follows:

1. First, we take $\beta = \eta$, and set

$$k = \Omega\left(\frac{\log 1/\eta^2\delta}{\eta\epsilon}\right).$$

   To be as in Lemma 2.

2. Divide $S$, the input sample, to $k$, disjoint datasets $S_1, \ldots, S_k$ of size $m$. Each data set, via $A$, defines a model $q^A_{S_i}$.

3. Next, we define the random sample

$$S_X = \{X_{q^A_{S_1}}, X_{q^A_{S_2}}, \ldots, X_{q^A_{S_k}}\} \in Z^K.$$

---

[1]which are all absolutely continuous w.r.t the uniform distribution

4. Apply the mechanism in Lemma 2 and output $a \in [0,1]^Z$ such that, w.p. $1 - \eta$, for all $z \in Z$:
$$|a(z) - \mathrm{freq}_{S_X}(z)| \leq \eta.$$

5. Let $p$ be any arbitrary distribution such that for every $z \in Z$:
$$|a(z) - p(z)| \leq \eta \tag{3}$$
(if no such distribution exists $p$ is any distribution). and output
$$q_S^B = p.$$

Notice that each sample $z_j$ affects only a single sub-sample $S_i$ and in turn only a single random variable $X_{q_{S_i}^A}$. The histogram function $a$ is then $(\epsilon, \delta)$-DP w.r.t to its input $S$. The output $p$, by processing is also private. We obtain, then, that the above algorithm is $(\epsilon, \delta)$-private.

We next set out to prove that $p = q_S^B$ is close in TV distance to $q_{S_A}^A$ in expectation. For ease of notation let us denote $X_i = X_{q_{S_i}^A}$. Notice that, with probability $(1 - \eta)$, for every $z$:
$$|a(z) - \mathrm{freq}_{S_X}(z)| \leq \eta,$$
in particular, there is a $p$ that satisfies the requirement in Item 5 (i.e. $\mathrm{freq}_{S_X}$ defines such a distribution) and Eq. (3) is satisfied. We then have that for every $z$:

$$\left| p(z) - \frac{1}{k} \sum \mathbf{1}[X_i = z] \right| \leq |p(z) - a(z)| + \left| a(z) - \frac{1}{k} \sum \mathbf{1}[X_i = z] \right|$$
$$\leq 2\eta. \tag{4}$$

We now move on to bound the total variation between the model $\mathbb{E}[q_S^B]$ and $q_{S_A}$, where expectation is taken over the randomness of $B$.

To show this, we will use the reverse inequality of the coupling Lemma, in particular if $(\hat{X}_B, \hat{X}_A)$ is a coupling of $q_S^B$ and $q_{S_A}^A$ (where $S$ and $S_A$ are now fixed), then:

$$\| \mathbb{E}[q_S^B] - q_{S_A}^A \| \leq \mathbb{P}(\hat{X}_B \neq \hat{X}_A). \tag{5}$$

Our coupling will work as follows, first we output $p = q_S^B$ and sample $\hat{X}_B \sim p$, and we let $\hat{X}_A = X_{q_{S_A}}$. This defines a coupling $(\hat{X}_B, \hat{X}_A)$. Applying Eq. (4), with $z = \hat{X}_A$, exploiting the fact that Eq. (4) holds with probability at least $1 - \eta$:

$$\mathbb{P}(\hat{X}_B \neq \hat{X}_A) \leq \frac{1}{k} \sum_{i=1}^{k} \mathbb{P}(X_i \neq X_{q_{S_A}}) + \eta$$
$$\leq 2\eta + \eta.$$

And we have that:
$$\mathbb{P}(\hat{X}_B \neq \hat{X}_A) \leq \frac{1}{k} \sum_{i=1}^{k} \mathbb{P}(X_i \neq X_{q_{S_A}}) + 3\eta \leq \frac{1}{k} \sum_{i=1}^{k} \frac{2\|q_{S_i}^A - q_{S_A}\|}{1 + \|q_{S_i}^A - q_{S_A}\|} + 3\eta.$$

And,
$$\mathbb{E}_{S_A, S} \| \mathbb{E}[q_S^B] - q_{S_A} \| \leq \mathbb{E}_{S_A, S} \frac{1}{k} \sum_{i=1}^{k} \left[ \frac{2\|q_{S_i}^A - q_{S_A}\|}{1 + \|q_{S_i}^A - q_{S_A}\|} \right] + 3\eta$$
$$\leq \mathbb{E}_{S_1, S_2 \sim S} \left[ \frac{2\|q_{S_1}^A - q_{S_2}\|}{1 + \|q_{S_1}^A - q_{S_2}\|} \right] + 3\eta$$
$$\leq \left[ \frac{2\,\mathbb{E}[\|q_{S_1}^A - q_{S_2}\|]}{1 + \mathbb{E}[\|q_{S_1}^A - q_{S_2}\|]} \right] + 3\eta \qquad \text{concavitiy of } \frac{2x}{1+x}$$
$$\leq \left[ \frac{2\alpha}{1 + \alpha} \right] + 3\eta \qquad \text{monotinicity } \frac{2x}{1+x}$$

