# OpenReview forum: "Can Copyright be Reduced to Privacy?"
_ICLR.cc/2024/Conference — Submitted to ICLR 2024_

### Official Review · Reviewer_ycKh · 2023-10-30

**Soundness:** 2 fair
**Presentation:** 2 fair
**Contribution:** 2 fair
**Rating:** 3
**Confidence:** 5

**Summary:**

The paper discusses the challenges and misfits of applying algorithmic stability approaches, such as differential privacy and near-access-freeness (NAF), to copyright disputes, with an emphasizes that while computer science methodologies, including algorithmic stability approaches, can assist policymakers in making more informed decisions, they may not be suitable for converting legal standards into rigid rules.

**Strengths:**

Strength:
1.	Discusses in details on the subtle differences between privacy and copyright
2.	provides a thorough and comprehensive analysis of the challenges and misfits associated with applying algorithmic stability approaches to copyright disputes.
3.	Critically evaluates the limitations and potential pitfalls of algorithmic stability approaches, such as differential privacy and near-access-freeness (NAF), in the context of copyright law.

**Weaknesses:**

Weakness:
1.	Overall, the technical contributions of this paper may seem weak. While I admit that the paper made contribution in capturing the differences between using existing algorithmic stability methods to determine copyright infringement and applying copyright law, the technical contributions is not significant and do not help to address the problems. The main theoretical result is Proposition 1, which most people would likely anticipate. From my opinion, the most valuable part is Section 3, but this part does not contain any technical discussion.

2.	Another issue with this paper is that critique of NAF and DP does not lead to new quantifiable algorithmic stability approach immediately, not does it shed light on how to modify these notions to better resonate with the essence of copyright law.

3.  The paper lacks empirical or experimental results that would solidly substantiate its technical claims. For instance, 1) DP does not allow high influence under a satisfactory 'copyright definition', 2) NAF can allow models that may memorize completely the training set as long as a content they output does not provide a proof for such memorization.

4. There is contradiction between the legal and technical discussions. For example, the authors initially assert that a satisfactory "copyright definition" must allow algorithms to be highly influenced. However, in discussing the 'overly exclusive' of algorithmic stability, they claim that protection is confined to concrete 'expressions' and does not encompass abstract "ideas".

5. While the paper conceptually identifies the problem, it does not propose a practical solution or methodology to prevent copyright infringement, nor does it provide any discussions towards that end.

**Questions:**

Questions:
1.	On page 2, the authors mentioned “We further propose a different approach to using quantified measures in copyright disputes, to better serve and reconcile copyright trade-offs.” However, I was not able to identify where in the paper this different approach is proposed.

2.	The title "Can copyright be reduced to privacy?" seems to be incoherent with the main subject matter and body of this paper. The entire discourse in Section 3 orbits around the disparity between practical application and algorithmic stability methods. These methods are not exclusively confined to Differential Privacy (DP), but are more associated with the more versatile concept of NAF. The main purpose of NAF is to provide a measurement of a model's copyright protection, rather than privacy protection. However, this discussion strays from the main theme of whether privacy can reduce to copyright. From my perspective, this paper is more centered on the incongruity between NAF and copyright protection in practice.

3. please provide some experimental results to further support your claims on the limitation of DP and the flexibility of NAF.

4.  clarify the contradictions between the legal and technical discussions, and provide empirical evidence if necessary.
5. discuss the potential solutions


3.	On page 9, I am still uncertain about the distinction between "convert murky standards into rigid rules" and "make legal standards less murky" in the context of computer science methodology. If we intend to incorporate CS methodology into the legal profession to clarify legal standards, rigid rules are inevitable. Could the authors further elaborate on this?

4.	The writing in some parts can be further improved. For example, what does \eta mean in Proposition 1.

---

> ### Author Response · Authors · 2023-11-21
> **Rebuttal**
>
> We thank the reviewer for his careful reading, below we address the weaknesses and concerns.
>
> **the technical contributions**
>
> The technical challenges behind our results are indeed not the main merit. Nevertheless, we do think that our results highlight that DP and NAF are more closely related than can be seen at first sight, and that both reduce to the requirement of a certain algorithmic stability that may be too prohibitive for the purpose of copyright infringement.
>
> **Another issue with this paper is that critique of NAF and DP does not lead to new quantifiable algorithmic stability approach immediately,**
>
> We acknowledge this as not just a limitation, but as an inherent aspect of the current discourse on copyright. We believe that underscoring the discrepancies between legal interpretations of copyright and its perceptions within DP and NAF frameworks is an important contribution and also demonstrates the complexity and potential limitations of developing a universally applicable, technical model. Our work, therefore, initiates a crucial dialogue, suggesting that existing formal notions may be flawed, and we highlight inherent issues. This perspective is essential for guiding future research and policy-making in a direction that acknowledges and accommodates these inherent complexities.
>
> **The paper lacks empirical or experimental results that would solidly substantiate its technical claims. For instance, 1) DP does not allow high influence under a satisfactory 'copyright definition', 2) NAF can allow models that may memorize completely the training set as long as a content they output does not provide a proof for such memorization.**
>
> + Regarding 1) We do not see how an experiment is in order here. As we discuss, there are works that can be blatantly non-private without infringing copyright. The experiment that the reviewer is proposing here would be something like running a generative AI exhaustively just to show that it will not generate a piece of art that parodies Shakespeare while preserving privacy of works by Shakespeare. That's not a realistic experiment, and we believe it is safe to say that a work that preserves privacy of A cannot parody or reference A.
>
> + Regarding 2) This statement is about the definition of NAF -- NAF, in principle, allows memorization. We do not claim that any realistic setup, or implementation of NAF, will necessarily lead to memorization.
>
> **There is a contradiction between the legal and technical discussions. For example, the authors initially assert that a satisfactory "copyright definition" must allow algorithms to be highly influenced. However, in discussing the 'overly exclusive' of algorithmic stability, they claim that protection is confined to concrete 'expressions' and does not encompass abstract "ideas".**
>
> We don’t see the contradiction here. In fact, the claims are complementary. A satisfactory definition of copyright must allow algorithms to derive “ideas” from previous work, but not to copy expressions. So some influence must be allowed, while other types of influences are not allowed.
>
> **On page 2, the authors mentioned “We further propose a different approach to using quantified measures in copyright disputes, to better serve and reconcile copyright trade-offs.” However, I was not able to identify where in the paper this different approach is proposed.**
>
> We provide a brief discussion at the final paragraph, but we will rephrase this statement as the reviewer is correct that we do not sufficiently discuss and delve into such an approach.
>
> **The title "Can copyright be reduced to privacy?" seems to be incoherent with the main subject matter and body of this paper. The entire discourse in Section 3 orbits around the disparity between practical application and algorithmic stability methods..**
>
> We indeed argue in the paper that the issue is not with privacy but with algorithmic stability in general. Our title alludes to recent attempts to tackle copyright through privacy and privacy-like notions, but the paper itself expands beyond that.
>
>
> **On page 9, I am still uncertain about the distinction between "convert murky standards into rigid rules" and "make legal standards less murky" in the context of computer science methodology. If we intend to incorporate CS methodology into the legal profession to clarify legal standards, rigid rules are inevitable. Could the authors further elaborate on this?**
>
> One potential contribution of CS methodologies to the law may be to provide evidence and quantitative information that could assist arbitrators in making informed decisions. That would still leave the judgement to a human that can assess and weight the pros and cons, without necessarily inflicting rigid rules.
>
> **The writing in some parts can be further improved. For example, what does $\eta$ mean in Proposition 1.**
>
> $\eta$ is the error parameter, and it affect the sample complexity. As two reviewers found this confusing, we will further clarify.

---

> > ### Comment · Reviewer_ycKh · 2023-11-22
> >
> > Thanks for your response. However, note that this is a computer science conference, no experiments and methods will heavily affect the contributions of this paper. So I will not change my score.

---

### Official Review · Reviewer_ttA1 · 2023-10-31

**Soundness:** 3 good
**Presentation:** 2 fair
**Contribution:** 3 good
**Rating:** 6
**Confidence:** 4

**Summary:**

This paper analyzes whether differential privacy (DP) and near access freeness (NAF) are sufficient to implement copyright law in ML models. DP is drawn from the privacy literature, and NAF is a recent copyright-inspired proposal that draws ideas from copyright to implement a kind of deliberate ignorance of specific training examples. The paper argues that some key notions in copyright, such as fair use and parody, are not adequately captured by DP/NAF.

**Strengths:**

The paper is legally informed and does a very good job applying legal doctrines. It includes a helpful discusison of recent caselaw and copyright scholarship, and the point made in section 4 about the limits of the NAF approach is very well-taken. The analysis in section 3 is particularly helpful in showing the ways in which these approaches can be both over and under-inclusive.

**Weaknesses:**

The writing is confusing. I had a hard time keeping straight over-inclusiveness, over-exclusiveness, under-exhaustiveness, stability, and safety. The paper could benefit from a careful pass to use consistent terminology.

I also think that the paper is discussing at least three kinds of issues: (1) whether there was copying in fact from a source work, (2) the quantitative degree of similiarity between an output and a source work, and (3) whether the use of a source work is justified in light of some approved legal purpose, such as parodic fair use or criticism. It is conceptually very difficult to separate (1) and (2), and the paper makes some attempts to clarify this line using DP/NAF, but I am not sure that it succeeds. On the other hand, (3) is very different and I am not sure that it is fair to critique NAF for excessive caution in saying that such examples have been copied. They have, and we currently lack computational tools to analyze whether the copying is justified.

**Questions:**

none

---

> ### Author Response · Authors · 2023-11-21
> **Rebuttal**
>
> Thank you very much for your feedback, below we address specific concerns:
>
> **The writing is confusing.**
>
> Thanks, we will improve this and use more consistent terminology. Indeed, there is no reason to use over/under inclusive/exclusive and it will be better to use unified terminology.
>
>
> **I also think that the paper is discussing at least three kinds of issues: (1) whether there was copying in fact from a source work, (2) the quantitative degree of similarity between an output and a source work, and (3) whether the use of a source work is justified in light of some approved legal purpose, such as parodic fair use or criticism. It is conceptually very difficult to separate (1) and (2), and the paper makes some attempts to clarify this line using DP/NAF, but I am not sure that it succeeds.**
>
> We are not trying to use DP/NAF in order to clarify the distinctive line between (1) and (2). We mainly critique the use of DP or NAF as measurements of such infringements. Partly, because it cannot provide such a quantitative degree of similarities
>
>
> **On the other hand, (3) is very different and I am not sure that it is fair to critique NAF for excessive caution in saying that such examples have been copied. They have, and we currently lack computational tools to analyze whether the copying is justified.**
>
> We agree that we lack such computational tools. Currently, it is a matter of the court to decide. But, as we argue in the paper, requiring NAF and DP instead, will indeed be too excessive, and it will not allow copying that is justified That being said, it does not mean that NAF or DP are not a key elements in providing future tools to assess copyright infringement. But they shouldn't be regarded as exhaustive tools.

---

> > ### Comment · Reviewer_ttA1 · 2023-11-22
> >
> > Thank you for this response and the commitment to clean up the terminology. I think we are talking past each other a bit on the other issues. I agree with you that DP/NAF cannot be used as-is to assess copyright infringement entirely algorithmically. But I think you are somewhat too dismissive of the possibility that they could be used *as part of* a copyright analysis that also includes human steps, and I think the paper could be improved by asking with greater precision which isue in a copyright case these tools bear on usefully, and which ones they cannot speak to.

---

### Official Review · Reviewer_rrnp · 2023-10-31

**Soundness:** 2 fair
**Presentation:** 1 poor
**Contribution:** 3 good
**Rating:** 3
**Confidence:** 2

**Summary:**

The authors examine two proposed computational measures, viz. differential privacy and Near Access Freeness, that have recently been proposed as measures that can be used to assess whether the generated content of a model counts as copyrighted content. The authors argue that neither of the measures constitute an adequate test of copyright infringement when considered within the context of U.S. legislation.

**Strengths:**

If correct, the authors conclusion significantly contributes to current research on assessing the copyright status of the generated content of a model. It shows that two recently proposed measures for this do not cohere with copyright as it is found in U.S. legislation. The comparison between the two disciplines is essential in developing computational measures that allow this assessment and, if correct, the discussion of the authors constitutes an original and important step in relation to this aim.

**Weaknesses:**

(1) The discussion is not sufficiently informed by the definition of copyright within U.S. legislation. It appeals to the goals of copyright legislation and to examples of what does and does not count as copyright according to the legislation, but not to the definition of copyright itself.
(2) In a few places, there is an inconsistency between claims. For example, on page two, the authors state that they will focus on challenges to providing a definition of copyright, while in Section 2, the focus is on algorithmic stability as a surrogate for copyright. These two are not the same, and will involve different implications and challenges. Possibly the claims can be reformulated in a clearer way that avoids the inconsistency.
(3) In many places, there are grammatical errors. For example, on page two, discussion is of Alice 'had she never saw' (vs seen) B.
(4) Unless an unfamiliar referencing convention is being used, the format for citation and referencing needs to be corrected throughout.

**Questions:**

(1) How does the definition of copyright as it is found within U.S. legislation relate to your discussion?

---

> ### Author Response · Authors · 2023-11-21
> **Rebuttal**
>
> We thank the reviewer for the comments and below we address the raised concerns and discuss how we intend to incorporate improvements.
>
> **The discussion is not sufficiently informed by the definition of copyright within U.S. legislation. It appeals to the goals of copyright legislation and to examples of what does and does not count as copyright according to the legislation, but not to the definition of copyright itself.**
>
> We do reference the statutory definition of copyright multiple times throughout the paper (U.S.C. 17 U.S.C. § 102(b). 2006.). But this is done implicitly and we do not emphasize it, and we should provide here more background.
>
> The “definition”, though, by itself, is an empty vessel. Like most complex legal concepts, it gains meaning only by court interpretation, which is the essence of this paper. We thank the reviewer for this comment, and below we provide a paragraph that will be incorporated in a future revision. This will help the reader to have better context:
>
> The statutory definition of Copyright is defined in Section 102 of the Copyright Act. Subsection 102(a), which at a high level says that almost all imaginable works are protected (“Copyright protection subsists… in original works of authorship fixed in any tangible medium of expression… include the following categories:
> 1) literary..
> 2) musical..
> 3) dramatic..
> 4) pantomimes..
> 5) pictorial..
> 6) motion pictures..
> 7) sound..
> 8) architectural..).
>
> For the purposes of our work, the definition of the **limitation** of copyright is the core-essence, and we reference in several occasions. Those are provided in in subsection 102(b), which defines the limits of copyrights :  (“In no case does copyright protection for an original work of authorship extend to any idea, procedure, process, system, method of operation, concept, principle, or discovery…).
>
> **In a few places, there is an inconsistency between claims. For example, on page two, the authors state that they will focus on challenges to providing a definition of copyright, while in Section 2, the focus is on algorithmic stability as a surrogate for copyright. These two are not the same, and will involve different implications and challenges.**
>
> Our intent is to explore the challenges in defining copyright, and in doing so, we examine existing approaches that rely on algorithmic stability (DP and NAF). We aim to illustrate why such approaches (that are based on stability) may not adequately address the needs of copyright. We hope this explains the alleged contradiction.
>
> **How does the definition of copyright as it is found within U.S. legislation relate to your discussion?**
>
> Please see our discussion above, and answer to (1). We hope it answers the question.

---

### Official Review · Reviewer_83w5 · 2023-11-07

**Soundness:** 2 fair
**Presentation:** 1 poor
**Contribution:** 2 fair
**Rating:** 3
**Confidence:** 2

**Summary:**

This article contends that it's crucial to recognize the distinctions between privacy and copyright. While algorithmic stability might seem like a useful method for identifying copying, it doesn't automatically ensure copyright protection. The authors pinpoint several discrepancies between algorithmic stability strategies and copyright regulations, illustrating why the implementation of such strategies might not fully consider fundamental copyright principles. Consequently, the paper suggests that if algorithmic stability techniques become the standard for addressing copyright infringement, they could undermine the original objectives of copyright law. The authors emphasize the necessity for any copyright concept to establish a clear distinction between protected expressions and unprotected ideas, highlighting a challenge that algorithmic stability concepts such as DP might not effectively address in certain cases. They provide instances where several elements of copyright law complicate the simplification of the copyright issue to a matter of privacy:
 - Copyright protections have a time limit, allowing works to enter the public domain after the expiration of protection. This implies that DP might be excessively stringent as a privacy notion.
 - Copyright law excludes certain subject matter from protection as they serve as raw material for cultural expression, a factor not readily addressed by privacy considerations.
 - Privacy protects content rather than expression, which differs from the scope of copyright law.
 - Copyright law promotes the use of copyrighted materials through specific transformative uses, including quotations and parodies, a dimension not fully encapsulated by privacy concerns.

In summary, the paper prompts critical thinking on why DP and NAF fail to effectively address the copyright problem. However, it lacks a comprehensive proposal for a practical resolution.

**Strengths:**

- The work provides an in-depth discussion on the relationship between DP, NAF and copyright law.
- The work studies a timely problem.

**Weaknesses:**

**No constructive copyright notion**. Authors look at special cases when DP or NAF would be too strict of a copyright notion and a relaxed notion could yield more benefit. While these special use cases make sense (e.g., Copyright is limited in time; copyright law encourages the use of copyrighted materials for transformative use cases) the authors merely identify these use cases without providing a technical copyright notion that would help make a step towards a more realistic copyright notion. In my view, both DP and NAF already do a good job at ensuring copyright law is upheld. As the authors note, to claim that the output of a generative model infringes copyright, a plaintiff must:
 1. prove that the model had access to her copyrighted work; and
 2. prove that the alleged copy is substantially similar to her original work.

Therefore, it is my understanding that any reasonable algorithm to ensure copyright, must either make sure that the model behaves as if it had no access to the copyrighted work or that the model makes sure that the generated output is dissimilar to the original work. In my understanding, (1) is guaranteed by DP and in doing that DP is quite conservative, e.g., copyright law does allow the output to be influenced by the original work if it is sufficiently dissimilar from the original work which DP prohibits. Another strategy can lie in simply making sure that the generated output is dissimilar to the original work. This is what NAF aims to achieve targeting point (2). To summarize, the work does not offer a constructive solution that helps solve these problems.

**Unclear theoretical results.** The theoretical results do not seem self-contained – e.g., to properly understand the paper, you must have read the work by Vyas et al (2023). Below are more concrete examples that are unclear:
  - “The premise in the above theorem is identical to that in Theorem 3.1 in Vyas et al. (2023)”: Does this mean that the full premise is stated in Proposition 1 or should the reader look at Theorem 3.1 of Vyas et al (2023) to understand the full premise of Proposition 1?
  - Some terms are not properly introduced: e.g., what is a “sharded-safety setting”?
  - Proposition 2 is presented in section 2; but the discussion on its implications is provided in Section 4.
  - In Proposition 1, eta is not defined. What is eta?

**Questions:**

See above.

Authors should consider using \citep instead of \citet (most of the time).

---

> ### Author Response · Authors · 2023-11-21
> **Rebuttal**
>
> We thank the reviewer for the comments and critique, we next address these and discuss how we intend to improve those points that require clarification.
>
> We do want to point out, however, that several of the points raised were already addressed in the paper itself. In addition, the reviewer has a different viewpoint and we respect that. But, we will focus, in our rebuttal, only on the content of the paper, and specifically on the arguments, as well as on supporting references and factual evidence, as they appear in the paper.
>
> Below is a detailed response.
>
> **No constructive copyright notion.**
>
> We acknowledge this as not just a limitation, but as an inherent aspect of the current discourse on copyright. We believe that underscoring the discrepancies between legal interpretations of copyright and its perceptions within DP and NAF frameworks is an important contribution and also demonstrates the complexity and potential limitations of developing a universally applicable, technical model. Our work, therefore, initiates a crucial dialogue, suggesting that existing formal notions may be flawed, and we highlight inherent issues. This perspective is essential for guiding future research and policy-making in a direction that acknowledges and accommodates these inherent complexities.
>
>
> **In my view, both DP and NAF already do a good job at ensuring copyright law is upheld.**
>
> Thank you for your perspective on the effectiveness of DP and NAF in upholding copyright law. Our paper, however, reaches a contrary conclusion: in our work, we systematically outline the limitations of using NAF and DP as criteria for determining copyright infringement, with our arguments supported by references and factual evidence. Should you feel that we have misrepresented any evidence, we are open to further discussion and committed to refining our analysis.
>
> We argue that the reliance on model access to copyrighted work and substantial similarity as the sole criteria for infringement is overly restrictive and may not align with the original intent of copyright law. Furthermore, our paper discusses why DP and NAF might not adequately safeguard these criteria. We look forward to a constructive dialogue on these issues and value your feedback in improving the depth and clarity of our work.
>
> **Therefore, it is my understanding that any reasonable algorithm to ensure copyright, must either make sure that the model behaves as if it had no access to the copyrighted work or that the model makes sure that the generated output is dissimilar to the original work.**
>
> The points you've raised are central to our paper and are extensively discussed within it. We welcome specific refutations or challenges to our arguments for a more detailed dialogue
>
> **Unclear theoretical results. The theoretical results do not seem self-contained – e.g., to properly understand the paper, you must have read the work by Vyas et al (2023). Below are more concrete examples that are unclear:**
>
> Thank you for your feedback. We'll enhance explanations, particularly about Vyas et al (2023), for better self-containment. Details on our additions are provided below
>
> + “The premise in the above theorem is identical to that in Theorem 3.1 in Vyas et al. (2023)”: Does this mean that the full premise is stated in Proposition 1 or should the reader look at Theorem 3.1 of Vyas et al (2023) to understand the full premise of Proposition 1?
>
> The full premise is stated in proposition 1, no need to look at theorem 3.1. We will clarify this. Thanks.
>
> + Some terms are not properly introduced: e.g., what is a “sharded-safety setting”?
>
> What we mean here is that NAF with respect to sharded-safe function can be satisfied. This part is indeed not self contained and we will elaborate and explain more here, thanks.
>
> + In Proposition 1, $\eta$ is not defined. What is $\eta$?
>
> The statement says that if the sample scales with $O(1/\eta)$ then the error scales with $\eta$. As this led to a confusion with a second reviewer we will emphasize here the role of $\eta$

---

### Meta-Review · Area_Chair_CfH7 · 2023-12-05

**Metareview:**

The paper undertakes an ambitious task of analyzing the relationship between algorithmic stability approaches like Differential Privacy and Near-Access-Freeness in the context of copyright law. It provides valuable insights into legal doctrines and copyright scholarship, highlighting significant conceptual differences between privacy and copyright. However, the paper falls short in several critical areas, necessitating its rejection. Firstly, the paper fails to offer a constructive copyright notion or a technical solution that addresses the identified issues. It acknowledges the limitations of DP and NAF but does not propose an alternative approach that resonates more closely with the essence of copyright law. Secondly, the theoretical results presented are unclear and not self-contained, requiring familiarity with other works for full comprehension. This lack of clarity extends to the paper's terminology, which is inconsistently used and confusing. The absence of empirical or experimental results to substantiate the technical claims further weakens the paper's contribution. Overall, while the paper identifies an important problem and conceptually contributes to the discussion, its lack of a practical solution, unclear theoretical foundation, and methodological shortcomings lead to its rejection.

**Justification For Why Not Higher Score:**

The paper is rejected due to its lack of practical solutions, unclear theoretical results, and methodological issues. The paper successfully identifies the gap between existing algorithmic stability methods and the requirements of copyright law but fails to propose a quantifiable or practical approach to bridge this gap. The lack of empirical evidence and clear, self-contained theoretical results further undermines its contribution.

**Justification For Why Not Lower Score:**

N/A

---

### Decision · Program_Chairs · 2024-01-16

Reject